# Alternative Splicing in Lung Adenocarcinoma: From Bench to Bedside

**DOI:** 10.3390/cancers17081329

**Published:** 2025-04-15

**Authors:** Wenjie Luo, Mingjing Xu, Nathalie Wong, Calvin Sze-Hang Ng

**Affiliations:** Department of Surgery, Prince of Wales Hospital, The Chinese University of Hong Kong, Hong Kong SAR, China; williamluo@surgery.cuhk.edu.hk (W.L.); mxu@surgery.cuhk.edu.hk (M.X.)

**Keywords:** lung adenocarcinoma, alternative splicing, translational medicine

## Abstract

Lung adenocarcinoma is the most prevalent pathological type of lung cancer. Alternative splicing generates multiple protein products from a single gene, significantly contributing to proteome diversity. Dysregulated splicing influences the development of lung adenocarcinoma and creates novel neoantigens for current immunotherapies, such as Chimeric Antigen Receptor T cell therapy. This review discusses alternative splicing events and therapeutic strategies, highlighting the potential to enhance anti-cancer treatments for lung adenocarcinoma.

## 1. Introduction

With an estimated 2.20 million new cases and 1.79 million deaths annually, lung cancer is one of the most common and lethal cancer types worldwide [1]. Two primary forms of lung cancer are non-small-cell lung cancer (NSCLC) and small-cell lung cancer, although NSCLC comprises approximately 85% of cases. Lung adenocarcinoma (LUAD) is the most prevalent pathological subtype of NSCLC and is characterized by high genomic heterogeneity [2,3,4].

Significant achievements in the molecular profiling for LUAD, the utilization of predictive biomarkers, and the advancement of therapies have led to considerable progress and improved outcomes for many patients [2,5,6]. Recently, the development of high-quality genome assemblies and recent innovations in long-read sequencing technology have elucidated the intracellular mechanisms of alternative splicing (AS) and facilitated comparisons of splicing profiles among closely related organisms, offering insights into the genetic complexity and oncogenicity of LUAD [7,8,9,10].

AS generates multiple polypeptides from a single gene that significantly enhances proteomic complexity and contributes towards diverse phenotypic traits [11,12]. Splicing aberrations have been implicated in the initiation and progression of various cancers, with advanced therapeutics targeting AS events emerging as novel anti-tumor strategies, including those for LUAD [13,14,15,16]. Notably, recent studies have illustrated the contribution of AS to the antigen repertoire of cancer cells and have demonstrated the potential for expanding immunotherapies that recognize novel AS-derived epitopes [17,18]. The putative neoantigens generated through AS serve as a substantial source for tumor-specific therapeutic targets [19,20,21].

In this review, we highlight aberrant splicing events as major drivers of LUAD and summarize therapeutic strategies targeting splicing alterations. Additionally, we explore the potential and perspectives of AS-based anti-tumor strategies, such as Chimeric Antigen Receptor (CAR) T cell therapy.

## 2. The Mechanism of AS in LUAD

AS is a complex mechanism underscoring eukaryotic gene expression that allows a single gene to produce multiple proteins. It is a process, by which different combinations of exons are joined together, or “spliced”, from a pre-mRNA transcript during RNA processing, contributing to the diversity and adaptability in protein structures and functions without increasing genome size [22]. For example, the epidermal growth factor receptor (EGFR) is a vital membrane protein that regulates cell signaling and growth, mediating proliferation and differentiation in LUAD [23,24]. According to the NCBI Gene database, EGFR has nine isoforms, a through i, each derived from distinct splicing mechanisms and exhibiting unique biological functions. All EGFR-AS isoforms present unique crystal protein structure predicted by AlphaFold 3.0 [25], as illustrated in Figure 1. Alterations of diverse functional domains lead to different amino acids (a.a.s), featuring diverse protein sizes and putative subcellular locations, as shown in Table 1.

According to the splice site usage, AS can be categorized into seven elementary types as follows: exon skipping (ES), intron retention, mutually exclusive exons (MX), alternative 5′ splice site and alternative 3′ splice site selection, and alternative first exon and alternative last exon splicing, as illustrated in Figure 2 [11]. We reviewed the splicing data profiled in TCGA SpliceSeq by the MD Anderson Cancer Center and summarized the AS events in LUAD [26], as shown in Figure 3. Among the various AS events investigated, ES emerged as the most predominant splicing pattern, whereas MX was identified as the least prevalent (Figure 3A). Notably, our distribution analysis quantified the number of genes harboring varying frequencies of AS events and revealed that 6677 out of a total of 10,314 genes (64.7%) exhibit multiple splicing events. An increasing number of AS events per gene results in a smaller number of genes (Figure 3B). These results underscore the intrinsic mechanisms of AS that enable LUAD cells to generate a wide diversity of protein isoforms from a limited number of genes.

Increasingly, studies and prognostic splicing models have been developed by unraveling the mechanism of AS and correlating splicing aberrations with the tumor microenvironment and immune heterogeneity in LUAD [27,28]. For example, a recent study constructed a novel AS-based prognosis signature that is associated with regulatory B cells and malignant pleural effusion in LUAD [29]. Furthermore, a multi-center study led by a research team from Switzerland delineates the pan-cancer AS landscape through a comprehensive analysis across tumors from 8705 patients. Among 32 types of cancer, LUAD is recognized as one of the most prominent tumors, characterized by elevated levels of aberrant AS events, particularly ES, which demonstrate an increase of over 30% in tumor samples compared to matched normal tissues. Regarding therapeutic implications, the study elucidates that LUAD exhibits a significant prevalence of tumor-specific exon-exon junctions, indicating an increased complexity of splicing mechanisms and the potential neoantigen generation [20]. The development of advanced strategies necessitates a deeper understanding of the natural splicing properties in LUAD.

## 3. Emerging Technologies to Assess AS in LUAD

In the past decade, RNA sequencing (RNA-seq) has emerged as an essential methodological advancement for the comprehensive transcriptome-wide analyses of differential gene expression and mRNA splicing [30,31]. A pivotal advantage of this technology is single-cell RNA-seq (scRNA-seq), which facilitates the evaluation of gene expression at the high resolution of individual cells, thereby enabling the identification of distinct cellular populations within LUAD [32,33,34]. Advanced scRNA-seq, integrated with quantitative profiling techniques of both short-read and long-read sequencings, enables researchers to illuminate the intricacies of transcript diversity and AS events that may exhibit dramatic variability among heterogeneous tumor cell subpopulations [35,36]. In recent studies, the application of single-cell sequencing has been instrumental in elucidating the metastatic dynamics and aberrant AS, yielding novel therapeutic targets for various malignancies, including acute myeloid leukemia [37], ovarian cancer [38], and LUAD [39,40].

Furthermore, emerging advancements in spatial transcriptomics extend the genomic analysis by preserving the spatial arrangement of LUAD architecture [34,41]. Intriguingly, recent studies have introduced spatial isoform transcriptomics (SITs), an explorative methodology designed to characterize spatial AS isoforms and sequence heterogeneity utilizing long-read sequencing. The innovative SITs enable the in situ capture of full-length transcripts across different tissue areas, simultaneously providing a comprehensive perspective on gene expression dynamics and AS landscapes [42,43]. Notably, spatial sequencing combined with RNA-seq has unveiled critical findings in LUAD, revealing that hMENA/hMENAΔv6 splicing isoforms exhibit elevated levels in cancer-associated fibroblasts within tumor tissues, alongside the high expression of stromal *FN1* and tertiary lymphoid structures (TLSs) that are preferentially localized in the peritumoral region. The dense expression of *FN1* may serve as a barrier to intertumoral TLS localization, in agreement with significant anti-tumor immune responses, and may affect the efficacy of immunotherapy through interactions with ILT3 [44,45].

Researchers are endeavoring to address the limitations inherent in individual technologies while enhancing their capacity to elucidate the biological functions of AS isoforms. ScRNA-seq, employing short-read assembly, offers robust sensitivity for transcript detection at the individual cell level, whereas spatial transcriptomics enables the contextual interrogation of gene expression patterns within tissues. Collectively, the incorporation of long-read sequencing augments analytical depth by delivering intricate details regarding AS events. Effectively, these sophisticated technologies facilitate a holistic approach to examining gene expression and the spatial dynamics of LUAD, ultimately leading to profound insights into the mechanisms of AS within the tumor microenvironment.

## 4. The AS Events Involving Driver Genes of LUAD

Emerging evidence suggests that LUAD primarily develops from a specialized subset of alveolar type II (AT2) cells [46,47,48]. The intrinsic disorder of AT2 cell self-renewal is mediated by signals transduced through the dysregulation of significant drivers, such as the EGFR–Kirsten rat sarcoma oncogene (KRAS) pathway, which is frequently hijacked during oncogenesis through aberrant AS [49,50]. A crucial factor in this process is the generation of splicing isoforms from major driver genes, which influence the initiation and progression of LUAD by altering normal cellular functions and pathways. A summary of LUAD-related AS isoforms is given in Table 2.

### 4.1. KRAS

As one of the most frequently altered oncogenes in LUAD, *KRAS* encodes a small GTPase that mediates the connection between growth factor signaling and several downstream signaling pathways [67,68,69], including the MAPK pathway [70]. The *KRAS* locus produces two variants, KRAS4A and KRAS4B, which result from the AS of the fourth exon, leading to distinct membrane-targeting sequences. Common activating mutations found in exons 1 or 2 confer oncogenic properties to both splice variants [71,72,73]. Additionally, the expression levels of KRAS4A and KRAS4B show significant variability across different tissues and developmental stages. A recent multi-institutional study reported that KRAS4A expression was significantly elevated in advanced-stage LUAD patients [71,74,75]. Previous research has demonstrated that the stable expression of KRAS4A in in vivo LUAD models significantly promoted proximal metastasis [51].

Intriguingly, the disruption of *KRAS* splicing switching to preferentially produce KRAS4A, mediated by the regulation of the DCAF15/RBM39 pathway, has been shown to inhibit cellular stemness during lung tumorigenesis, presenting a potential therapeutic strategy for modulating *KRAS* splicing [76,77]. Furthermore, an exploratory analysis revealed that PD-L1 levels were significantly higher in patients with elevated KRAS isoforms, particularly those with high levels of KRAS4A [78]. Given these findings, KRAS4A merits further investigation as a potential biomarker for identifying patients who may benefit from immune checkpoint inhibitor therapy, thus enhancing personalized cancer immunotherapy.

### 4.2. EGFR

As a member of the receptor tyrosine kinase family, EGFR plays a crucial role in regulating cell growth and differentiation in response to its natural ligands, such as EGF and TGF-α [79,80,81]. Upon ligand binding, EGFR undergoes dimerization and autophosphorylation, which activates downstream signaling pathways [82,83,84], including the RAS-RAF-MAPK [85] and PI3K-AKT pathways [83,86]. Recently, mutations in *EGFR*, particularly within the tyrosine kinase domain, have been identified as vital drivers of LUAD tumorigenesis and have been associated with predicted sensitivity to tyrosine kinase inhibitors (TKIs) [87,88,89]. The most prevalent mutations, such as exon 19 deletion and the L858R point mutation in exon 21, lead to the constitutive activation of the receptor, which enhances tumor proliferation, survival, metastasis, and other cancer-associated properties [90,91].

Additionally, the AS of the *EGFR* gene can generate various isoforms that may significantly alter the receptor’s function and therapeutic responses, particularly to TKIs. The most common splicing variant, EGFRvIII, is caused by the truncation of exons 2–7 and leads to the deletion of a portion of the extracellular domain [92,93,94]. This variant exhibits unique constitutive activity independent of ligand binding, thereby contributing to TKI resistance in lung tumorigenesis [53]. Despite these insights, the regulatory mechanisms governing EGFR splicing in LUAD remain poorly understood and necessitate further investigation to elucidate their impact on tumor biology and therapeutic outcomes.

### 4.3. C-Met

The activation of the receptor tyrosine kinase, c-Met, has been implicated in the promotion of various cancers via the stimulation of multiple downstream oncogenic signaling pathways, including the MAPK, PI3K/AKT, and STAT pathways [95,96,97]. Aberrant c-Met signaling can facilitate lung tumorigenesis through a range of upstream mechanisms, such as *c-Met* gene amplification, mutations, rearrangements, and overexpression [98,99,100]. Recently, the exclusion of c-Met exon 14 (designated as MET-ΔEx14) [55] has been identified as a prototypical aberrant splicing event that exhibits oncogenic properties and holds considerable clinical significance in LUAD [101,102,103]. The exclusion of exon 14 leads to an in-frame deletion of 47 amino acids within the juxtamembrane domain of c-Met, which results in inhibited degradation and the internalization of the receptor [104]. Consequently, this alteration is classified as a gain-of-function event. The juxtamembrane domain functions as a crucial negative regulatory region of c-Met, containing a caspase-cleavage sequence and a tyrosine-binding site, which mediates the ubiquitination and subsequent degradation of c-Met [105,106].

Although MET-ΔEx14 occurs in approximately 4% of patients with LUAD [6,107,108], multiple studies have reported on MET-ΔEx14 as an independent oncogenic driver and a biomarker significantly associated with reduced survival rates in these patients [56,109,110]. Furthermore, growing clinical evidence indicates that patients harboring MET-ΔEx14 could benefit from MET TKIs, including crizotinib [111], tepotinib [112,113], and capmatinib [114]. Intriguingly, despite a substantial proportion of LUAD patients with MET-ΔEx14 alteration expressing PD-L1 [115], they often demonstrate a lower overall response rate to immune checkpoint blockade (ICB) therapies and experience shorter median progression-free survival [116].

### 4.4. CD44

CD44 is a glycoprotein that plays a critical role in cell–cell and cell–matrix interactions. According to the UniProt database, CD44 naturally exists as a diverse array of at least 20 different transmembrane isoforms [117]. Such structural heterogeneity is primarily attributed to AS [118]. The CD44 protein is encoded by a single gene consisting of 20 exons, 10 of which are subject to AS. Exons 1 to 5 and exons 16 to 20 are consistently expressed across various cell types. In contrast, exons 6 to 15 (designated as v1–v10) undergo AS, leading to the generation of numerous variant isoforms.

The protein variability of CD44 contributes to the functional heterogeneity in various biological processes [119,120,121], including cancer cell stemness and migration [122,123]. In LUAD, specific CD44 splicing patterns have been associated with tumor grade, stage, and patient survival [124,125,126]. Notably, the reduced expressions of CD44-v3 and -v6 have been linked to LUAD invasion [127]. CD44 AS variants could also influence the efficacy of anti-tumor immunotherapies, such as CAR T [58] and CAR NK therapy [128].

### 4.5. PD-1/PD-L1

Targeting the immune checkpoint molecules PD-1 and PD-L1 through ICB has shown significant clinical responses across various cancer types, including lung cancer [129,130], leukemia [131], and lymphoma [132,133]. Anti-PD-1/PD-L1 immunotherapy disrupts the PD-1/PD-L1 signaling pathway [134], which restores T cell activity, enhances anti-tumor immunity, and ultimately contributes to the elimination of cancer cells [135,136].

Despite these advancements, only a subset of patients respond to ICB, underscoring the need for a deeper understanding of the mechanism driving immune escape [59,137]. Research has revealed that exon 3 skipping in PD-1 leads to the generation of a soluble isoform, sPD-1, which lacks a transmembrane domain [138,139]. This soluble variant may antagonize PD-1, potentially enhancing anti-tumor immunity by interfering with the PD-1/PD-L1 signaling axis [140]. A clinical investigation in Denmark indicated that elevated sPD-1 expression is associated with improved outcomes for patients with *EGFR*-mutant NSCLC receiving the EGFR TKI erlotinib [141]. Moreover, high levels of sPD-1 expression have been linked to increased response rates to immunotherapies, such as anti-PD-1 and anti-CTLA4 therapies in NSCLC, including LUAD [142].

Additionally, a novel spliced variant of PD-L1, referred to as sPD-L1, which also lacks the transmembrane domain, has been identified in NSCLC [143]. The significance of pretreatment with ICB monotherapy for NSCLC patients harboring both sPD-1 and sPD-L1 has been emphasized [144]. Intriguingly, findings suggest that AS can generate a long non-coding RNA from the *PD-L1* gene, which promotes the progression of LUAD by enhancing c-Myc activity [145,146]. Collectively, recent evidence highlights the intricate role of AS of PD-1 and PD-L1 in immune regulation and tumor progression, suggesting potential avenues for improving the efficacy of immunotherapy in LUAD.

## 5. Therapeutic Strategies Targeting AS in LUAD

Aberrant AS has emerged as a significant regulatory mechanism that diversifies the functional repertoire of proteins involved in lung tumorigenesis, thereby contributing to the genetic complexity and heterogeneity in LUAD. To enhance the therapeutic efficacy, recent research underscores the potential in targeting specific splicing events or the splicing machinery in LUAD. This positions AS as an essential component in future personalized cancer treatment, paving the way for innovative approaches to therapy.

### 5.1. Small-Molecule Inhibitors

The therapeutic strategy of small-molecule inhibitors acting on splicing machinery and regulatory elements is believed to effectively correct abnormal splicing events. By redirecting oncogenic spliced variants to non-oncogenic ones, there exists the potential to restore normal cellular functions and counteract malignancy in LUAD [147,148].

Recent studies have identified nonsynonymous mutations in genes encoding members of the spliceosome complex, such as *SF3b1* [149], *U2AF1* [150], and *SRSF2* [151] in the development of LUAD. H3B-8800 is a small-molecule inhibitor that inducing preferential lethality in spliceosome-mutant epithelial tumor cells, such as LUAD [152]. Mechanistically, H3B-8800 specifically interacts with the SF3b complex, a critical component of the spliceosome complex within the branchpoint binding that occurs during splicing [153]. Intriguingly, the competitive binding of H3B-8800 to mutant SF3b1, which is a critical member of SF3b complexes in its canonical form and induced aberrant splicing events, leads to dose-independent inhibition through the dissociation of the mutant spliceosome [152,153,154]. Moreover, a recent study combined molecular dynamics simulation trajectories with metadynamics simulations to identify the existence of a putative druggable SF3b pocket in the vicinity of the SF3b1-K700E mutation, aiming to develop future therapeutic strategies that specifically target the mutant spliceosome while sparing the wild-type spliceosome [154]. Additionally, H3B-8800 has shown promising results in preclinical studies, demonstrating anti-tumor activity across various spliceosome-mutant cancer models, including LUAD [152]. A phase I clinical trial identified a subset of patients with acute myeloid leukemia who could benefit from H3B-8800 treatment, indicating an acceptable adverse effect profile [155]. Currently, clinical evaluations of H3B-8800 in LUAD remain underdeveloped and resistance to its inhibition has also been observed, underscoring the complexity of targeting splicing factor mutations [152,156,157].

Another spliceosome inhibitor is E7107, which specifically targets the U2 snRNP complex in pre-mRNA splicing [158,159]. Similarly to H3B-8800, E7107 exerts its effects through competitive binding to the mutant SF3b complex [160], effectively blocking spliceosome assembly and thereby activating multiple pro-apoptotic signaling pathways, such as the p53 pathway [161] and the regulation of BCL2 family [162]. Recent clinical trials have underscored its anti-tumor efficacy across a range of solid tumors, including LUAD [163,164]. Certain spliceosome inhibitors, including Spliceostatin A [165] and Pladienolide [166], modulate the fidelity of branch point selection by binding competitively to the SF3b complex, underscoring their potential as therapeutic agents for innovative anti-splicing strategies in LUAD. In conclusion, the potent anti-tumor effects of H3B-8800 and E7107, these two derivatives of pladienolide B [167], demonstrate the therapeutic potential of spliceosome inhibitors in addressing oncogenic splicing dysregulation in LUAD. The functional mechanisms of spliceosome inhibitors are illustrated in Figure 4.

Beyond the spliceosome, small-molecule inhibitors targeting specific splicing regulatory proteins have also demonstrated significant therapeutic potential. These compounds aim to disrupt the function of key splicing factors, thereby altering splicing patterns that drive tumorigenesis.

Protein arginine methyltransferase 5 (PRMT5) is a splicing regulator that plays a critical role in pre-mRNA splicing by methylating spliceosomal proteins. JNJ-64619178, inhibitor of PRMT5, leads to the inhibition of symmetric arginine dimethylation of SMD1/3 proteins, core components of the spliceosome in the tumor and alters aberrant splicing by suppressing PRMT5 enzymatic activity [168,169]. Xenograft models of LUAD have demonstrated that JNJ-64619178 effectively inhibits tumor growth and promotes regression, making it a promising candidate for splicing-targeted therapies [170].

Another key splicing regulator, RBM39, has been targeted by an aryl sulfonamide drug, indisulam [171]. Indisulam functions by recruiting DCAF15, which bridges RBM39 with the E3 ubiquitin ligase complex [172,173]. This action leads to the polyubiquitination and subsequent proteasomal degradation of RBM39, thus disrupting RBM39-dependent pre-mRNA splicing [172]. Intriguingly, a separate preclinical study indicates that targeting the DCAF15/RBM39 axis with Indisulam modulates the differential splicing of KRAS 4A/4B, which is essential for the development of *KRAS* mutant LUAD. Indisulam significantly inhibits initiation of LUAD by suppressing stem cell-like properties, positioning it as a cutting-edge therapeutic strategy against this malignancy [76].

Furthermore, small-molecule inhibitors have been actively pursued to target specific splicing isoforms, with one of the most representative examples being MET-ΔEx14 [98]. Small molecule MET TKIs such as crizotinib and cabozantinib, demonstrate potent activity against MET amplification thorough lower nanomolar potency in MET-ΔEx14 mutated cell lines [174]. A study conducted by Memorial Sloan Kettering Cancer Center enrolled Stage IV LUAD patients harboring exon 14 alterations, who exhibited significant radiographic responses to treatment with crizotinib and cabozantinib [175]. These results encourage prospective identification of MET exon 14 splicing alteration in LUAD patients.

In conclusion, small-molecule inhibitors directly disrupt the complex splicing machinery or isoforms within cancer cells, causing the mis-splicing of numerous RNA transcripts. This alteration can lead to selective lethality for tumor cells that rely on aberrant AS for their survival, underscoring the potential of splicing modulation in LUAD therapy.

### 5.2. RNA-Targeted Therapies

RNA-targeted therapies, including RNA interference (RNAi) and antisense oligonucleotides (ASOs) [176], aim to silence critical targets, such as KRAS and EGFR, by redirecting AS in LUAD metastasis [177,178].

A complementary RNAi strategy targeting KRAS has identified TBK1-activated NF-κB anti-apoptotic signals that involve c-Rel and the splicing of Bcl-xL. Both c-Rel and spliced Bcl-xL are essential for LUAD proliferation and thus provide mechanistic insights into this synthetic lethal interaction [179]. Furthermore, AZD4785, a high-affinity ASO targeting KRAS, can potently and selectively deplete cellular KRAS mRNA and protein resultant in the inhibition of downstream effector pathways and antiproliferative effects, especially in *KRAS*-mutant LUAD cells [52].

ASOs targeting EGFR are designed to induce exon skipping within the transmembrane or tyrosine kinase domains that can modulate the production of specific splice variants associated with LUAD proliferation and migration [180,181]. Wang et al. synthesized and introduced the ASO VF/S/A@CaP, which sensitizes osimertinib-resistant LUAD cells, characterized by a high mesenchymal state, to ferroptosis [182]. Recent findings indicate that EGFR-targeting ASO exhibits a more potent anti-cancer effect than traditional TKIs in NSCLC with *EGFR* mutations, including L858R and T790M, and can effectively suppress TKI-resistant patient-derived tumors [54]. By influencing EGFR splicing, these ASOs aim to alter receptor expression or function in the cell membrane, thus potentially reducing tumor growth and improving responses to existing therapies.

Recent studies showed that the strategy of splice-switching antisense oligonucleotides (SSOs) has high therapeutic efficiency [183,184]. SSOs hybridize pre-mRNA sequences and modulate AS by exposing or masking the exon–intron boundaries. The administration of nanoparticles containing Bcl-x SSOs, which interfere with the splicing pattern of Bcl-x pre-mRNA, exhibit in vivo anti-tumor activity in LUAD models [185]. A schematic presentation of RNA-targeted therapies is illustrated in Figure 5.

### 5.3. Emerging Gene Therapies

RNA targeted therapies have emerged as a formidable strategy for silencing specific oncogenes in LUAD. Despite its promise, the clinical applicability of them is hindered by the transient nature of its effects and the obstacles associated with effective delivery mechanisms [186,187]. To overcome these limitations, researchers have introduced innovative gene therapies utilizing engineered oncolytic viruses (OVs) that preferentially enter cancer cells by exploiting receptors that are overexpressed in tumors, while sparing healthy tissue and leveraging their natural properties to proliferate [188]. These OVs not only induce direct lysis of tumor cells but also provoke systemic anti-tumor immune responses through the release of tumor-associated antigens [189]. Notable OVs include Coxsackievirus B3 [190], Reolysin [191], and NDV-rL-RVG [192], each demonstrating promising results in LUAD treatment.

Recently, Yun et al. introduced an oncolytic adenovirus (OAs)-based short hairpin RNA (shRNA) expression system, specifically targeting vascular endothelial growth factor (VEGF) through the Ad-ΔE1-shVEGF construct. Among the seven isoforms of VEGF, the VEGF-121 sequence is conserved across all isoforms, allowing VEGF-121-specific shRNA to effectively degrade all seven VEGF variants. To achieve efficient and sustained VEGF silencing, the study incorporated the shVEGF construct under the regulation of a U6 promoter within the E3 region of a double mutant OAs characterized by E1A and E1B deletions. The modified OAs exhibited substantial tumor-suppressive effects in LUAD A549 cell lines in vitro [193]. Furthermore, research suggests that the inhibition of the specific isoform VEGF-165, a prominent splicing isoform of VEGF, can elicit long-term anti-tumor responses when integrated into standard oncolytic virus therapy regimens [194]. Such targeted inhibition tailors the complexities inherent in AS, which generates multiple VEGF isoforms with distinct biological functions. This underscores a pivotal intersection between oncolytic virotherapy and splicing modulation, offering a novel avenue for enhancing therapeutic efficacy against LUAD.

Furthermore, CRISPR/Cas9 gene editing provides a complementary strategy for gene therapies through its capacity for precise genomic alterations [195]. This technology allows for the disruption of oncogenes or the restoration of tumor suppressor genes in LUAD [196]. Recent studies utilized CRISPR/Cas9 to knock out the m6A-modified SNRPA, which further induced exon 8 skipping of the *ERCC1* gene, thereby reversing the SNRPA-mediated cisplatin resistance observed in LUAD [197]. Additionally, researchers have developed pgFARM (paired guide RNAs for alternative exon removal), a CRISPR/Cas9-based methodology aimed at manipulating isoforms without necessitating gene inactivation using LUAD xenografts [198]. This approach employs a pooled screening technique to assess the functional significance of “poison” cassette exons, which disrupt the reading frames of their host genes while demonstrating a high degree of evolutionary conservation.

The emergence of gene therapy technologies, encompassing OVs therapy and CRISPR/Cas9 gene editing, offers promising new strategies for the treatment of LUAD. Continued exploration and research in this area are essential for unlocking the full potential of these strategies by investigating the intricate mechanisms of AS in LUAD.

### 5.4. Combining Splicing-Targeted Strategies with Standard Treatments

In recent decades, treatment of LUAD has experienced significant advancements in targeted therapies, particularly through the use of TKIs to target specific driver mutations such as *EGFR*, *ALK*, and *ROS1* [199,200,201]. However, resistance to these therapies due to a combination of genetic alterations, cellular modifications, and microenvironmental factors remains a prevalent challenge [202,203]. Intriguingly, recent research has revealed that the dysregulated splicing of cancer-associated genes, such as *HER2* [204,205] and *BIM* [206,207], can be implicated in mediating acquired TKI resistance in LUAD.

Vorinostat is an FDA-approved, orally taken, histone deacetylase inhibitor drug for primary T-cell lymphoma [208]. Required resistance to apoptosis induced by TKI osimertinib in LUAD can be overcome by the combined use of vorinostat, which affects the AS of BIM mRNA and increases the expression of active BIM protein [209]. This offers an alternative off-label use of vorinostat. Moreover, a clinical trial has indicated that the combination of vorinostat and gefitinib can be effective for patients with BIM-deletion polymorphisms and *EGFR* mutations [210].

Recent studies have shown that the co-occurring deficiency of the splicing factor RBM10 in *EGFR*-mutant LUAD can contribute to EGFR inhibitor osimertinib resistance [211,212]. This resistance is associated with a decreased apoptotic response resulting from splicing alterations in Bcl-xL, which is intrinsically modulated by RBM10 [213]. Importantly, researchers have found that the combination of a Bcl-xL-specific ASO and osimertinib could effectively overcome this resistance, leading to the enhanced killing of LUAD [62].

Targeting aberrant splicing in conjunction with conventional treatment options, such as TKIs, represents a promising strategy to combat LUAD. By addressing the AS-derived molecular vulnerabilities, clinicians aim to enhance therapeutic efficacy to further improve LUAD patient outcomes.

### 5.5. Upcoming AS-Based Immunotherapies

Immunotherapy has marked a significant advancement in cancer treatment by harnessing the immune system to recognize and combat cancer cells, including LUAD [214,215]. A noteworthy strategy in this field is adoptive cell therapy using CAR T cells, which is one of the antibody-based anti-tumor therapies and considered the most promising personalized immunotherapeutic treatment [216]. A notable success in this area is the CD19-targeted CAR T cell therapy, which has been approved for the treatment of relapsed or refractory acute lymphoblastic leukemia [217,218]. Although CAR T cell therapy has achieved remarkable success in treating hematological malignancies, efforts are underway to extend its application to solid tumors. Research is rapidly advancing to assess the feasibility and effectiveness of CAR T cell therapy for LUAD [219]. The identification of optimal tumor antigens is crucial for developing targeted CAR T immunotherapies [18,216].

Alternative mRNA processing in LUAD has been shown to alter proteomic diversity, providing an extensive but largely unexplored repertoire of novel immunotherapy targets [220,221]. Recent studies highlight the utility of a novel system called isoform peptides from RNA splicing for immunotherapy target screening (IRIS) [17]. IRIS utilizes a computational platform that is capable of identifying AS-derived tumor antigens, thereby broadening the scope of cancer immunotherapies. Among those upcoming tumor-specific targets, the CD44 AS isoform CD44v6 has been implicated in LUAD tumorigenesis, invasion, and metastasis [57,222]. Anti-CD44v6 CAR T cells have demonstrated functional activation in vitro, with the ability to infiltrate tumors, proliferate, inhibit LUAD growth in vivo, and improve overall survival in mouse models [58]. The interplay between LUAD cells and anti-CD44v6 CAR-engineered T cells is illustrated in Figure 6.

A substantial class of splicing-derived tumor-specific neojunctions highlights the potential to utilize AS-based proteins for diagnosis, prognosis, and therapeutic interventions [21]. As research advances, integrating these insights into immunotherapy strategies offers promises for developing more effective and personalized anti-LUAD treatments.

## 6. Conclusions

This review presents the multifaceted role of AS in the pathogenesis and treatment of LUAD. High-throughput screening methodologies, coupled with advanced bioinformatics analysis, are crucial for the comprehensive identification of novel AS events and associated biomarkers. This integration enables more precise patient stratification and the selection of appropriate targeted therapies, ultimately leading to improved treatment outcomes. Simultaneously, a deeper understanding of the complex regulatory mechanisms governing AS in LUAD is paramount, which necessitates the exploration of the intricate interplay between splicing machinery and the LUAD microenvironment. Integrating emerging technologies such as scRNA-seq and spatial transcriptomics enhances the elucidation of AS expression dynamics and spatial interactions.

Emerging therapeutic strategies aim to directly target AS machineries or specific aberrant splicing events. Small-molecule inhibitors and RNA-targeted therapies offer novel approaches to correct aberrant splicing and potentially overcome therapeutic resistance. The advent of gene therapy technologies, including engineered OVs and CRISPR/Cas9 gene editing, enriches the landscape of AS-modulating therapies for LUAD. Combinatory therapies, including the refinement and optimization of existing technologies, such as integrating AS-targeted ASOs with TKIs for patients harboring *EGFR* mutation, are able to enhance specificity and efficacy in LUAD while mitigating off-target effects. Furthermore, rigorous preclinical evaluations would prioritize the efficacy and safety of new drugs, and the addition of robust AS-derived biomarkers would further enable identification of LUAD patients who would most likely benefit from AS-targeted therapies.

In sum, the dynamic interaction between AS and the immune microenvironment plays an essential role in LUAD. The concerted effort to identify and characterize novel AS-derived neoantigens represents an unprecedented opportunity to advance cancer immunotherapies, such as CAR T cell therapy. The transition from basic research findings to clinical treatments allows us to bridge the gap from bench to bedside, and this hinges on continued AS-based investigations into LUAD.

## Figures and Tables

**Figure 1 cancers-17-01329-f001:**
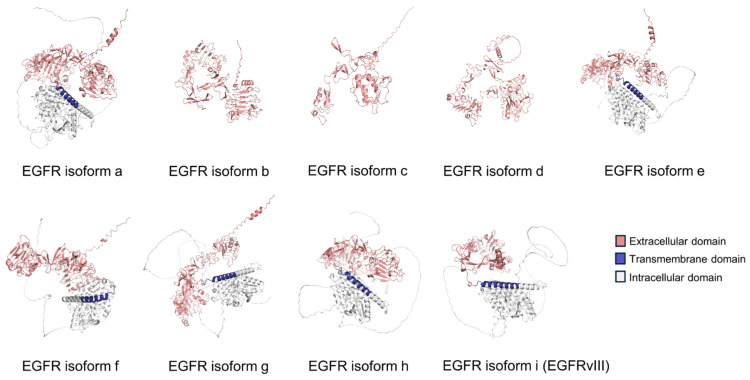
Predicted protein structures of EGFR isoforms using PyMOL 2.6 and AlphaFold 3.0.

**Figure 2 cancers-17-01329-f002:**
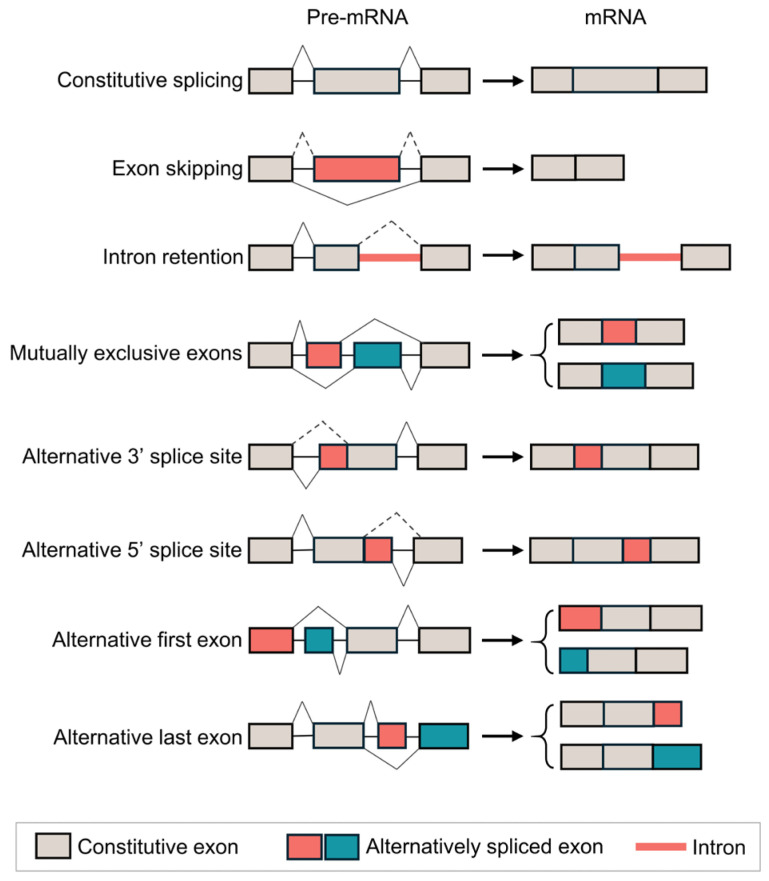
Constitutive splicing (top) and seven elementary types of alternative splicing (created with Microsoft PowerPoint 2024).

**Figure 3 cancers-17-01329-f003:**
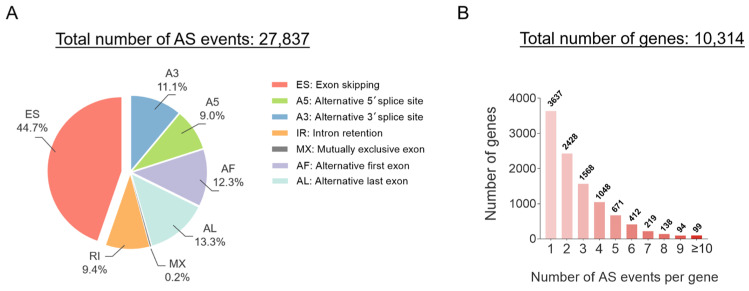
Summary of AS events in LUAD derived from TCGA SpliceSeq. (**A**) Proportion of different types of AS events in LUAD. (**B**) Distribution analysis of a total of 10,314 genes characterized by the number of AS events per gene. A percentage of samples with Percent Spliced In (PSI) values greater than 75% was included in analysis. (Created with GraphPad Prism 8.0 and Microsoft PowerPoint 2024).

**Figure 4 cancers-17-01329-f004:**
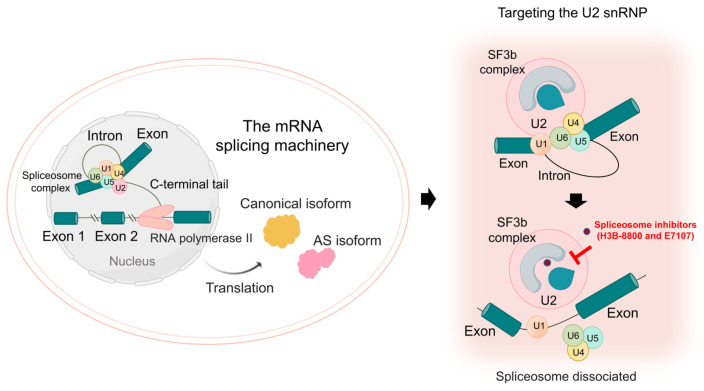
Small-molecule inhibitors, such as H3B-8800 and E7107, inhibit the spliceosome through direct interaction with the SF3b complex (created with Microsoft PowerPoint 2024 and Adobe Illustrator 2025).

**Figure 5 cancers-17-01329-f005:**
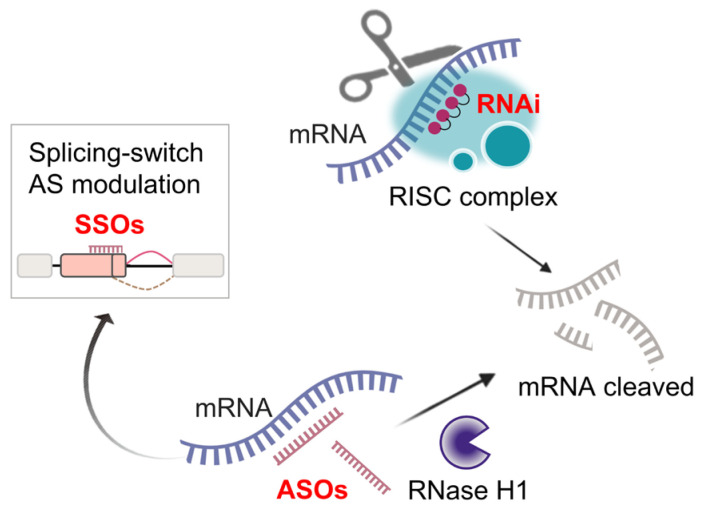
Overview of RNA-targeted therapies, including RNAi, ASOs, and SSOs (created with Microsoft PowerPoint 2024 and Adobe Illustrator 2025).

**Figure 6 cancers-17-01329-f006:**
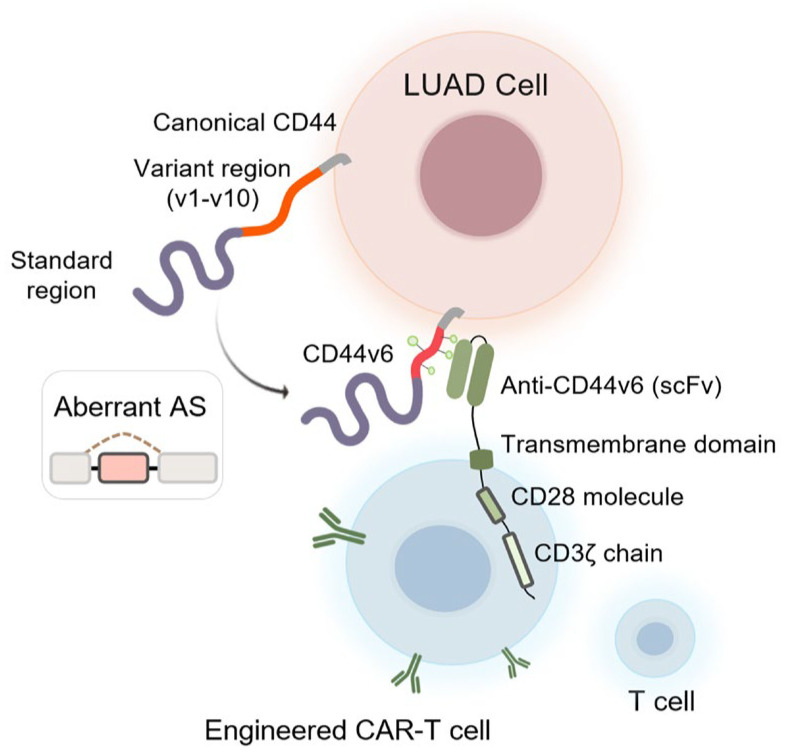
Tumor-specific CD44v6 derived from aberrant AS represents a promising neoantigen for CAR T cell immunotherapy in LUAD (created with Microsoft PowerPoint 2024 and Adobe Illustrator 2025).

**Table 1 cancers-17-01329-t001:** Detailed characteristics of EGFR AS isoforms in the NCBI Gene database.

EGFR AS Isoform	AS Type	Number of a.a.s	Protein Size (kDa)	Predicted Subcellular Location
Isoform a	Canonical isoform	1210	134.3	Cell surface protein
Isoform b	Alternative last exon	628	69.2	Secreted protein
Isoform c	Alternative last exon	405	44.7	Secreted protein
Isoform d	Alternative last exon	705	77.3	Secreted protein
Isoform e	Exon skipping and alternative last exon	1091	120.7	Cell surface protein
Isoform f	Alternative last exon	1136	125.8	Cell surface protein
Isoform g	Exon skipping	1165	129.2	Cell surface protein
Isoform h	Alternative first exon	1157	128.7	Cell surface protein
Isoform I (EGFRvIII)	Alternative first exon	943	104.3	Cell surface protein

**Table 2 cancers-17-01329-t002:** LUAD-related AS isoforms.

Gene	AS Event	Biological or Clinical Implications	AS-Related Therapy	Reference(s)
*KRAS*	Inclusion of either exon 4A (KRAS4A) or exon 4B (KRAS4B)	KRAS4A induces metastatic LUAD in vivo	ASO	[51,52]
*EGFR*	Deletion of exons 2–7 (EGFRvIII)	EGFRvIII contributes to resistance against TKIs	ASO	[53,54]
*c-Met*	Deletion of exon 14 (MET-ΔEx14)	MET-ΔEx14 confers clinical sensitivity to MET inhibitors	Small-molecule inhibitors	[55,56]
*CD44*	Inclusion of variant exon 6 (CD44v6)	CD44v6 is implicated in the process of lung tumorigenesis	CAR T cell therapy	[57,58]
*PD-1*	Deletion of exon 3 (sPD-1)	sPD-1 enhances anti-tumor immunity	Anti-PD-1/PD-L1 therapy	[59,60]
*Bcl-x*	Alternative last exon (Bcl-xL)	Bcl-xL inhibits tumor cell apoptosis	ASO	[61,62]
*CDC25C*	Exon 3 skipping (CDC25C-ΔEx3)	CDC25C-ΔEx3 inhibits cell proliferation	/	[63]
*KAT2A*	Alternative 5′ splicing	The splicing variant promotes LUAD progression	/	[64]
*EIF4H*	Exon 5 skipping (EIF4H-ΔEx5)	EIF4H-ΔEx5 promotes LUAD progression	/	[65]
*IMPAD1*	Alternative last exon	The splicing variant promotes LUAD proliferation	/	[66]

*C-Met*, c-mesenchymal–epithelial transition factor; *PD-1*, programmed cell death 1; *PD-L1*, programmed cell death ligand 1; *Bcl-x(L)*, B-cell lymphoma extra-large; *CDC25C*, cell division cycle 25C; *KAT2A*, lysine acetyltransferase 2A; *EIF4H*, eukaryotic translation initiation factor 4H; *IMPAD1*, inositol monophosphatase domain-containing protein 1.

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
