# Peer review of "Alternative Splicing in Lung Adenocarcinoma: From Bench to Bedside"

_cancers, 2025, doi:10.3390/cancers17081329_

Round 1
Reviewer 1 Report
Comments and Suggestions for Authors
The review article is well-written and the flow of the paper is very good. My major concern is that you have not mentioned whether the figures were generated or copied from elsewhere. Either, please mention the App utilized to generate the figures or reference the source of the figures.
Minor changes:
Start of line 202: Please change 'attributed by AS' to 'attribute to AS'.
Line 219: Please change 'deeper understanding on the mechanism' to 'deeper understanding of the mechanism'.
Reviewer 2 Report
Comments and Suggestions for Authors
While the topic of alternative splicing (AS) in lung adenocarcinoma (LUAD) is indeed of interest, the review lacks sufficient novelty and critical analysis. The main concerns are as follows:
The review largely compiles existing knowledge on alternative splicing (AS) in lung adenocarcinoma (LUAD) without offering new perspectives or significant updates.
Many of the referenced studies are well-known and have been extensively cited in prior reviews.
There is no clear effort to synthesize new hypotheses, propose novel mechanisms, or integrate emerging technologies (e.g., spatial transcriptomics, single-cell RNA-seq).
While the review provides a general overview of AS mechanisms, it does not delve deeply into molecular regulation, such as the specific role of RNA-binding proteins or how mutations in spliceosomal components contribute to LUAD pathogenesis.
Some sections repeat basic textbook knowledge about AS without connecting it to specific LUAD phenotypes.
The review does not critically evaluate conflicting data or highlight controversies in the field.
There is no discussion of potential limitations in current research methodologies or gaps that need to be addressed.
The therapeutic strategies mentioned (e.g., ASOs, small molecule inhibitors) are described in a generic manner without critical assessment of their clinical viability or current challenges.
Several sections reiterate the same points using different wording, particularly regarding the impact of AS on immune evasion and therapy resistance.
Figures and tables seem to repackage information that is already well-documented, without providing unique insights.
The review does not sufficiently discuss emerging research directions or propose innovative approaches for targeting AS in LUAD.
There is minimal discussion on how multi-omics approaches, AI-based splicing predictions, or personalized medicine could improve the field.
Comments on the Quality of English LanguageSubstantial language revision is needed.
Reviewer 3 Report
Comments and Suggestions for Authors
In this review article, Luo et al. nicely summarize alternative splicing in lung adenocarcinoma. They discuss the alternative splicing events and therapeutic strategies, highlighting the potential to enhance anti-cancer treatments for lung adenocarcinoma.
This is a well written review and very informative to the readers of IJMS. I have a few suggestions and comments described below.
1) It would be nicer if the authors could prepare schematic representations of alternative splicing of each gene in Figures like Figure 1.
2) As for SSA, Dr. Valcarcel group published the papers and they found that SSA affects fidelity of branch point selection, in addition to inhibition of spliceosome assembly (doi: 10.1101/gad.2014311.). This should include and discussed in the text.